# Prevalence of physical health conditions and health risk behaviours in people with severe mental illness in South Asia: protocol for a cross-sectional study (IMPACT SMI survey)

Gerardo A Zavala ![ORCID] ,[1] Krishna Prasad-Muliyala,[2] Faiza Aslam,[3] Deepa Barua,[4] Asiful Haidar,[4] Catherine Hewitt,[1] Rumana Huque,[4] Sonia Mansoor,[3] Pratima Murthy,[2] Asad T Nizami,[3] Najma Siddiqi,[1,5,6] Siham Sikander,[7] Kamran Siddiqi ![ORCID] ,[1,6] Jan Rasmus Boehnke ![ORCID] ,[1,8] On behalf of the IMPACT team

For numbered affiliations see end of article.

**Correspondence to**
Dr Gerardo A Zavala;
gzavala@gmail.com

## ABSTRACT

**Introduction** People with severe mental illness (SMI) die on average 10–20 years earlier than the general population. Most of these deaths are due to physical health conditions. The aim of this cross-sectional study is to determine the prevalence of physical health conditions and their associations with health-risk behaviours, health-related quality of life and various demographic, behavioural, cognitive, psychological and social variables in people with SMI attending specialist mental health facilities in South Asia.

**Methods and analysis** We will conduct a survey of patients with SMI attending specialist mental health facilities in Bangladesh, India and Pakistan (n=4500). Diagnosis of SMI will be confirmed using the Mini-international neuropsychiatric interview V.6.0. We will collect information about physical health and related health-risk behaviours (WHO STEPwise approach to Surveillance (STEPS)); severity of common mental disorders (Patient Health Questionnaire-9 (PHQ-9) and General Anxiety Disorder scale (GAD-7)) and health-related quality of life (EQ-5D-5L). We will measure blood pressure, height, weight and waist circumference according to WHO guidelines. We will also measure glycated haemoglobin, lipid profile, thyroid function, liver function, creatinine and haemoglobin. Prevalence rates of physical health conditions and health-risk behaviours will be presented and compared with the WHO STEPS survey findings in the general population. Regression analyses will explore the association between health-risk behaviours, mental and physical health conditions.

**Ethics and dissemination** The study has been approved by the ethics committees of the Department of Health Sciences University of York (UK), Centre for Injury Prevention and Rehabilitation (Bangladesh), Health Ministry Screening Committee and Indian Council of Medical Research (India) and National Bioethics Committee (Pakistan). Findings will be disseminated in peer-reviewed articles, in local and international conferences and as reports for policymakers and stakeholders in the countries involved.

### Strengths and limitations of this study

► The study uses standardised tools to measure patient outcomes that will allow us to compare the findings with those of the general population.
► The survey will be conducted face to face by trained researchers.
► The questionnaire has been translated into the most common languages used in Bangladesh, India and Pakistan using validated translations of standardised measures where available.
► The survey is being conducted in specialist centres and is therefore a healthcare utilisation sample drawn largely from tertiary care.
► The study population may not be representative of all the people with severe mental illness in each country, as those not accessing specialist centres are not included in the survey.

**Trial registration number** ISRCTN88485933; 3 June 2019.

## BACKGROUND

A considerable body of research has shown that people with severe mental illness (SMI; that is, psychotic disorders, bipolar affective disorder and severe depression with psychotic symptoms) die on average 10–20 years earlier than the general population.[1 2] Around 80% of deaths in people with SMI are due to preventable physical illnesses, most commonly cardiometabolic diseases, respiratory disorders and infectious diseases.[3 4] This excess mortality is a major global public health challenge but efforts to address it have been limited. Recent studies suggest that despite an overall improvement in life expectancy for both the general population and people with

SMI, the absolute mortality gap between these two groups is actually widening.[5–7]

Prevalence of most physical health conditions is higher and outcomes are poorer in this population.[8] Multimorbidity (the presence of two or more conditions) is also more common[9] in the presence of SMI, contributing to poorer physical health and quality of life. The majority of evidence for these health inequalities has been generated in high-income countries (HICs), but a small number of studies from low-income and middle-income countries (LMICs) also show a similar pattern of increased mortality for people with SMI, with an even shorter life expectancy, and larger mortality gap.[4 10–12] The few studies available suggest that physical comorbidity in SMI is at least as prevalent as in developed countries.[1]

In South Asia, rates of mental illness and physical non-communicable diseases have been increasing rapidly.[13 14] This increase, coupled with the lack of access to even basic mental healthcare,[15] and neglect of the physical health needs of people with SMI by policymakers and healthcare services,[16] means that the burden of disease due to physical disorders in people with mental illness. is set to rise further, with a corresponding increase within country and global health inequalities. This increase is coupled with a recognised lack of information about the prevalence of physical conditions and health risk behaviours in the SMI population in South Asia.[17] This gap in knowledge, particularly in relation to health service planning in LMICs, has been pointed out repeatedly as a priority for research.[18] Addressing mental and physical health comorbidity in LMICs is also a global priority, recognised in global policies to help achieve the sustainable development goals, including 'ensuring healthy lives and promoting well-being for all'.[19–21] A detailed understanding of the prevalence of physical health conditions, health risk behaviours and health service utilisation in SMI in LMICs is needed to progress this agenda.[22]

### Objectives
In people with SMI attending specialist mental health facilities:
1. Determine the prevalence of physical health conditions (ie, type 2 diabetes, hypertension, heart disease, obesity and infectious diseases) and health-risk behaviours (ie, diet, physical activity, alcohol and tobacco use) and compare these findings with those reported in the general population.
2. Determine the association between physical health conditions, health-risk behaviours, health-related quality of life and various demographic, behavioural, cognitive, psychological and social variables.
3. Identify health-service utilisation and the type of lifestyle advice that has been offered for this population.
4. Support clinical trial readiness by providing evidence-based findings regarding outcome measures and procedures relevant for clinical trial design.

## METHODS AND ANALYSIS
### Design, settings and population
We will conduct a cross-sectional survey among patients with a clinical diagnosis of SMI recruited at national specialist mental health institutions. In Bangladesh, the study will be conducted at the National Institute of Mental Health and Hospital (NIMHH) in Dhaka. NIMHH offers 200 beds for in-patient care and on average 400 outpatients attend every day. In India, the study will be conducted at the National Institute of Mental Health and Neurosciences (NIMHANS) in Bengaluru, with 650 beds for in-patient care; on average at NIMHANS, 400 outpatients attend every day. In Pakistan, the study will be conducted at the Institute of Psychiatry (IOP) Rawalpindi Medical University, a WHO collaborating centre for the Eastern Mediterranean . IOP offers 60 beds for inpatients and on average 400 outpatients attend every day. All three national institutes serve patients from all over the country. Adults (18 years and over) with a clinical diagnosis of SMI (defined by the International Classification of Disease 10th revision (ICD-10[23]) as schizophrenia, schizotypal and delusional disorders (F20-F29); bipolar affective disorder (F30, F31) and severe depression with psychotic symptoms (F32.3, F33.3)),[23] who are able to provide informed consent as assessed by the treating clinician will be recruited. Patients who lack capacity to provide consent or are unable to complete study questionnaires will be excluded (figure 1). The data will be collected from June 2019 to September 2020.

### Sample size
We aim to build as large a sample as possible within the resources available over the study period (n=1500 per country). As an indicative example of survey precision to address some of the key research questions, we provide a sample size estimate for investigating the prevalence of diabetes. Assuming a prevalence estimate of 10% (based on results from previous studies,[24] we will need 857 participants per country to achieve ±2% precision assuming a 95% CI.[25]

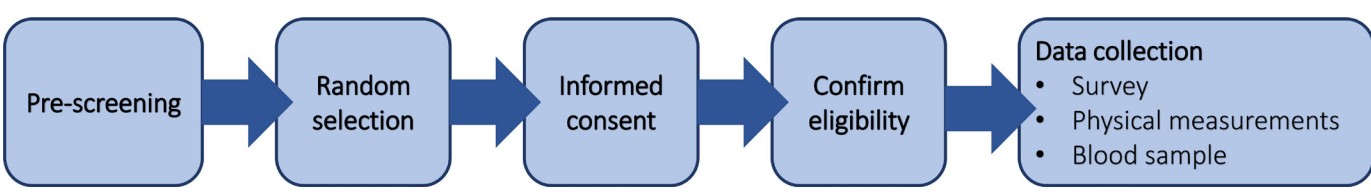

**Figure 1** Study flowchart.

## Recruitment and random selection of the participants

Both inpatients and outpatients attending one of the three specialist mental health institutions collaborating in this study will be recruited. At NIMHH and NIMHANS, patients will be selected randomly using random number tables generated centrally to determine for outpatients, which patient to approach on a given day, and for inpatients by randomly selecting beds. Recruitment is separated by inpatients versus outpatients, resulting in a proportion of 80% outpatients and 20% inpatients (which has been reported as the 'usual' proportion of inpatients and outpatients). Due to the lower volume of patients and outpatients reaching the centre outside working hours at the IOP, all patients attending the IOP during the study duration will be invited to participate in the survey.

## Informed consent

In the first instance, the researchers will give an information sheet to the patient, providing written and verbal information on the potential benefits and risks of taking part in the study and clarifying the assessments involved in the study. Only people who provided written informed consent will be included in the study. Where there are problems with literacy, the researchers will provide this information verbally to both the carer and the patient. Those unable to provide a signature will be requested to indicate their consent with a thumbprint. The information sheet will meet all the requirements of the relevant ethics committees. Maintenance of confidentiality and compliance with international, UK and relevant local Data Protection Acts will be emphasised to all study participants. Moreover, all participants will be informed that their participation in the study is voluntary, consent can be withdrawn at any stage and the decision about participation will not affect their care.

Mental capacity is time specific and decision specific. For patients who are assessed to lack capacity by their local physician, seeking informed consent will be delayed until capacity is regained, if feasible. No assessments will be continued where the patient appears reluctant, even where patient consent has been obtained.

Optional 'consent to contact' about future studies will be sought from survey participants. Permission will be sought to retain participant details and to contact participants about future studies that may be relevant for them. Again, it will be emphasised that this is voluntary, they can withdraw permission at any time and this will not affect their care in any way. After consenting to participate, the participant will be given a unique patient identifier (ID) by the researcher. The patient consent and contact information will be entered into a database (participant log), which will be kept separate from the survey result database. The survey result database will include only the unique patient ID and no patient identifiable data.

## Confirmation of eligibility

SMI diagnosis will be confirmed by trained researchers using the relevant modules from the Mini-International Neuropsychiatric Interview (MINI) V.6.0.[26] The MINI is a short diagnostic structured interview to explore mental disorders. It is designed to allow administration by non-specialist interviewers.[27] The inter-rater reliability for the MINI will be assessed at each site at the start of the study. Within each study site, at least 10 MINI questionnaires will be completed and compared between two different researchers blinded from each other. Where needed, further training will be offered to achieve acceptable inter-rater reliability before researchers are assigned to the study.

## Data collection

We will conduct a face-to-face digital survey to collect information about physical health and health-risk behaviours, severity of common mental disorders and health-related quality of life using validated instruments as described below. The survey has been translated into the most common languages in the countries (Urdu in Pakistan, Bangla in Bangladesh and Hindi, Kannada, Tamil and Telugu in India). The team of interviewers in each county will include men and women and will include researchers who speak regional dialects, which is consistent with usual clinical practice in these settings.[28] Participants will be able to express a preference to be interviewed by a researcher of the same gender, to reduce response bias that might arise due to the gender of the interviewer.

The WHO STEPwise approach to Surveillance (STEPS) instrument V. 3.2 will be used to collect information about non-communicable diseases and related health-risk behaviours.[29] STEPS is an international standardised tool that will allow comparisons with the general population within the country and between countries.[30] The instrument has already been translated, used and validated in the general population in Bangladesh, India and Pakistan.[31–33]

To allow patients to disclose information on sexually transmitted diseases, HIV diagnosis, alcohol and tobacco consumption, in agreement with patient's wishes, the interviewer will ask the caregivers/attendants to leave the room for this part of the interview.

### Demographic information

The STEPS demographic module will be used to collect information on age, sex, education, marital status, employment, household assets and income.[29]

### Mental health

We will collect information relevant to the SMI diagnosis of each participant, including duration of the mental illness, any treatment or medication and duration of the treatment and/or medication. The 9-item depression module (Patient Health Questionnaire-9 (PHQ-9))[34] will be used to measure the severity of depressive symptoms and the generalised anxiety disorder 7-item scale (GAD-7)[35] for the severity of anxiety symptoms.

### Health risk behaviours

The STEPS tobacco modules will be used to assess tobacco-related behaviours.[29]

### Alcohol consumption

The STEPS alcohol module will be used to determine lifetime abstainers, past 12 months abstainers and current users of alcohol using the WHO cut-off scores.[29]

### Diet

The STEPS diet module will be used to record the number of days that respondents consumed fruit and vegetables in a typical week, and the number of servings of fruit and vegetables consumed on average per day.[36]

### Physical activity

The STEPS physical activity module will be used to record activity for transport purposes (eg, walking, cycling), vigorous and moderate activity at work and vigorous and moderate activity in leisure time and time spent sitting. Show cards with culturally relevant examples will be used to aid respondents in classifying activities. Analysis and categorisation will follow the WHO guidelines.[37]

### Non-communicable diseases

Any medically diagnosed history of raised blood pressure (BP), heart disease, hypercholesterolemia and high blood glucose, and treatment advised by a health worker for these diseases (such as prescribed medicines, a special diet, advice to reduce salt intake, lose weight, stop smoking or do more exercise) will be collected using the STEPS module for non-communicable diseases. Lung disease is not included in the STEPSs survey. However, it will be recorded asking the same set of questions as the non-communicable diseases included in the STEPS survey.

### Infectious diseases

Self-report of medically diagnosed history of hepatitis B, C, syphilis, tuberculosis, HIV and intestinal parasites will be recorded.

### Risk behaviours related to sexually transmitted diseases

Risk behaviours related to sexually transmitted diseases will be assessed using three questions that examine the presence of multiple sexual partners, unprotected sexual contact and use of injectable drugs that have been adapted from the 10-item HIV risk Screening Instrument.[38] This instrument has been previously used in research involving persons with SMI in India.[39]

### Health-related quality of life

The EQ-5D-5L will be used to measure health-related quality of life.[40 41] The EQ-5D-5L is a standardised measure of health status, it provides a simple, generic measure of health for clinical and economic appraisal. The impact of disease/disorder is characterised on five dimensions (mobility, self-care, ability to undertake usual activities, pain and anxiety/depression). The EQ-5D-5L further assesses a patient's subjective evaluation of their health state based on a visual analogue scale (EQ-5D-VAS) between '0—worst imaginable health state' and '100—best imaginable health state'. We will use validated translations provided by EurOQol.

### BP and heart rate

BP and pulse rate will be taken three times using an automated BP measuring instrument (OMRON). For BP measurement, the participant must be comfortable and rested. If he/she had been exerting themselves, then there will be a minimum of a 15 min rest period before the recording is taken. There should be at least a 3 min gap between the BP recordings. The procedure will follow the stepwise instructions in the WHO STEPS surveillance manual.[42]

### Height weight and waist circumference

Height will be measured to a precision of 0.1 cm using a portable height measuring board without footwear and headgear. Weight will be measured in kilograms using a portable weighing scale placed on a firm flat surface with light clothing and without footwear and socks. Waist circumference will be measured in a precision of 0.1 cm, using a flexible fibreglass anthropometric tape at the end of normal expiration, at the midpoint between the lower margin of the last palpable rib and the top of the iliac crest (hip bone), with the arms relaxed at the sides. All measurements will be taken in duplicate and the average of the two values will be recorded and the protocols of the WHO STEPS surveillance manual will be followed.[42]

### Biochemical analysis

A blood sample (8–9 mL) will be taken from consenting participants for glycated haemoglobin (HbA1c), lipid profile (triglycerides, total cholesterol, high-density lipoproteins (HDL), low-density lipoproteins (LDL)), thyroid function tests (includes thyroid hormones (T3, T4), thyroid stimulating hormone (TSH)), liver function tests (total bilirubin, aspartate transaminase (AST), alanine aminotransferase (ALT), alkaline phosphatase (ALP), total protein, albumin, albumin to globulin ratio), creatinine and haemoglobin.

### Data management

To ensure quality of collected data, validation, checking, proofing and cleaning procedures will be carried out according to standard procedures.[43] All consent forms will be stored separately from survey data in locked cabinets in locked offices at study research offices at each study site. All coded data will be transferred to and stored as anonymous data at the University of York, which will act as the data curator. A secure password protected and encrypted electronic database will be set up to store the data.

### Statistical analysis

Quantitative data will be summarised using descriptive statistics. First, prevalence rates of infectious diseases,

long-term physical health conditions (eg, diabetes, heart disease) and health-risk behaviours for type II diabetes, cardiovascular and respiratory disorders (eg, tobacco use, obesity, physical activity, diet) will be estimated in the SMI population. Second, we will examine the associations between physical health conditions and health-risk behaviours, using regression models with random intercepts for data collection site with the following potential covariates: demographic variables (ie, age, sex and level of education), history of mental and physical conditions and biomarkers (eg, body mass index, BP). These regression models will also be used to calculate prevalence rates of physical health conditions and health-risk behaviours adjusted for these covariates (eg, for comparison with external general population data). Further regression analyses will explore the association between mental and physical health conditions and health-related quality of life measured using EQ-5D-5L. Regression models will take account of the nature and distribution of data (eg, count vs continuous data; skewed vs normally distributed data). To investigate the prevalence of physical health conditions and health risk behaviours in the general population, we will extract aggregated data on the prevalence of physical health conditions, health risk behaviours and lifestyle advice from the last available WHO STEPS survey from Bangladesh,[44] India[45] and Pakistan.[46]

## ETHICS AND DISSEMINATION

The study has been approved by the ethics committees of the Department of Health Sciences University of York (UK), Centre for Injury Prevention and Research (Bangladesh), Health Ministry Screening Committee and Indian Council of Medical Research (India) and National Bioethics Committee (Pakistan).

The study will adhere to the fundamental principles of human rights and dignity laid down in the Declaration of Helsinki.[47] Study procedures will comply with legislation and guidance for good practice governing the participation in research of people lacking capacity as set out in the Mental Capacity Act (UK) 2005.[48] Participants will not receive any financial inducement to participate, but the results of the physical measurements and blood tests will be shared with the participant and the treating clinician.

As a non-interventional study, there is minimal risk of adverse events associated with the study. Due to the vulnerability of the SMI population, however, there is a potential (although low) risk that some questions or the burden of the assessments may cause distress for participants or the family carers. If there is any indication of this, the survey or assessments will be stopped, the participant offered reassurance and support, and if needed, referral to the clinical team. If a participant during the interview reveals any suicidal ideation, the interviewer will refer them to the treating clinician for a clinical risk assessment and further management. If a physical condition or blood test abnormality (outside the normal range for age and sex) is detected as a part of the research assessments, the research team will inform the clinician responsible for the patient, who will evaluate the result and give the patient the option of referral to a specialist if this is required for their condition.

## Community, patient and public involvement

A 'Community Panel' comprising stakeholders from these constituencies ensures community, patient and public involvement. The panel has reviewed the planned survey and advised on its feasibility. The panel will continue to provide feedback on the conduct of the study and presentation of findings. To support dissemination efforts, findings will be presented to the community panel, with advice sought about the format and content of lay summaries and other outputs aimed at patients and the public.

## Dissemination

Findings will be disseminated in peer-reviewed articles, in local and international conferences and as reports for policymakers and stakeholders in the countries involved. An open access repository of all materials will be built that covers the survey materials as well as statistical syntax used by the team publications.

## DISCUSSION

People with SMI have a reduced life expectancy. Research in HICs has shown that a large proportion of this inequality is accounted for by chronic diseases.[1 2] This has been consistently observed in HICs.[3 4] However, information is lacking in minority groups and LMICs, which is surprising considering the increased risk of physical health conditions in these populations due to poverty, education, poor diet, low access to healthcare, exposure to infectious diseases and other environmental factors.[49] This survey will address this gap in knowledge, demonstrating the prevalence of different physical health conditions and health-risk behaviours in the SMI population and the association of each of these conditions in terms of health-related quality of life. It will support SMI clinical trial readiness by providing evidence-based findings regarding outcome measures and procedures relevant for clinical trial design.

### Strengths and limitations

To the best of the authors' knowledge, this will be the first large-scale study to investigate the physical health and health-risk behaviours in the SMI population in LMICs. The main limitation is that the survey is conducted in specialist centres (with patients with capacity) and is therefore a healthcare utilisation sample drawn largely from tertiary care. The gaps in treatment and care utilisation and systematic factors influencing these have been well documented,[50] therefore, the study population may not be representative of the total SMI population in each country. However, a community survey would not be feasible, and primary and secondary care

mental health services are not sufficiently developed to be used to recruit participants for this survey (see, eg, work by PRIME[15] and Emerald[51] programmes on strengthening these services as access points). However, unlike services in HICs, tertiary care services in these South Asian countries accept self-referral without the need for primary or secondary care referral and we include outpatient services offered by the collaborating centres which often function as 'the first port of call' for people with mental illness.[52–54] Therefore, the selected centres are likely covering a broad range of the treatment-seeking population. We will not have a control group since the study is only conducted in people with SMI, however we are using the same questions and measurements as the WHO STEPS survey. This will allow us to compare the prevalence of physical health conditions and health risk behaviours with that of a survey representative for the population in each country.

**Author affiliations**
[1]Department of Health Sciences, University of York, York, UK
[2]Department of Psychiatry, National Institute of Mental Health and Neurosciences (NIMHANS), Bangaluru, India
[3]Institute of Psychiatry (IOP), Rawalpindi Medical University, Rawalpindi, Pakistan
[4]ARK Foundation, Dhaka, Bangladesh
[5]Bradford District Care NHS Foundation Trust, Bradford, UK
[6]Hull York Medical School, York, UK
[7]Global Health Department, Health Services Academy, Islamabad, Pakistan
[8]School of Health Sciences, University of Dundee, Dundee, UK

**Acknowledgements** We would like to acknowledge Dr Sue Bellass, Prof. Santosh Kumar Chaturvedi, Dr Arun Kandasamy, Dr Noreen Mdege, Dr Masuma Mishu, Dr Emily Peckham, Prof. Simon Gilbody and the research teams at ARK foundation and The National Institute of Mental Health (NIMH) in Bangladesh, Institute of Psychiatry in Pakistan (IOP) and The National Institute of Mental Health and Neuro-Sciences (NIMHANS) in India for their contribution to the study. We want to thank all the participants and caregivers for taking the time to complete the survey.

**Contributors** NS conceived the study, KP-M, FA, DB, RH, SM, PM, ATN, SS and KS provided important contextual information, KP-M, FA, DB, AH, SM, NS, KS and JRB developed the methodology and selected the outcomes, GAZ, AH, CH and JRB developed the analysis plan, GAZ, NS, KS and JRB wrote the manuscript, and all authors revised and approved the manuscript.

**Funding** This research was funded by the National Institute for Health Research (NIHR) (17/63/130) using UK aid from the UK Government to support global health research. The views expressed in this publication are those of the author(s) and not necessarily those of the NIHR or the UK Department of Health and Social Care. The funders had no role in study design, data collection and analysis, decision to publish, or preparation of the manuscript.

**Competing interests** None declared.

**Patient and public involvement** Patients and/or the public were involved in the design, or conduct, or reporting or dissemination plans of this research. Refer to the Methods section for further details.

**Patient consent for publication** Not required.

**Provenance and peer review** Not commissioned; externally peer reviewed.

**ORCID iDs**
Gerardo A Zavala http://orcid.org/0000-0002-9825-8725
Kamran Siddiqi http://orcid.org/0000-0003-1529-7778
Jan Rasmus Boehnke http://orcid.org/0000-0003-0249-1870

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
