## [Reviewer comments · BMJ Open]

ARTICLE DETAILS

TITLE (PROVISIONAL)	Prevalence of physical health conditions and health risk behaviours in people with severe mental illness in South Asia: Protocol for a cross-sectional study (IMPACT SMI survey)
AUTHORS	Zavala, Gerardo A.; Prasad, Krishna; Aslam, Faiza; Barua, Deepa; Haidar, Asiful; Hewitt, Catherine; Huque, Rumana; Mansoor, Sonia; Murthy, Pratima; Nizami, Asad; Siddiqi, Najma; Sikander, Siham; Siddiqi, Kamran; Boehnke, Jan Rasmus

VERSION 1 – REVIEW

REVIEWER	Rebecca Bendayan King's College London United Kingdom
REVIEW RETURNED	07-Apr-2020

GENERAL COMMENTS	This study tackles a relevant and timely need for research and focuses on a population traditionally neglected. The objectives are well aligned with the background and although clear they are still a bit broad and additional specifications in this section and through the protocol would be helpful to promote reproducible research. Design, settings and population: It would be helpful if the authors could specify which diagnostic criterion are they using to define their cohort. For example, it would be helpful to have the specific ICD codes for the SMI diagnosis. Moreover, age range and other potential socio-demographic factors for inclusion and exclusion would be interesting to have. For the sample size, it seems that there would be a number of participants by country. Are the authors going to use a stratified sampling strategy? As within institutions there will be also some randomization, would they follow a multi-stage strategy? With regards to the data collection, what is the plan to address the variability of dialects in countries as Pakistan and inequalities derived from urban vs rural areas, or potential stigma related to the topic. Some of these interviews might will need a local translator which can compromise confidentiality and the representativity of the sample. Is there any contingency plan for this? Also there could be some response bias related to the gender of interviewers or respondents (e.g., women will be less likely to respond some questions to male interviewers). Overall, all the information collected seem to be aligned with the objectives of the study but I find that two potential key elements are missed or not reported -medication and socio-demographics (e.g., neighbourhood level, social disadvantage)- which have been consistently found to be relevant confounders in the associations that the authors are interested. Finally, more details in the statistical plans would be helpful to promote replications and ensure comparability.
--

REVIEWER	christian ritter Northeast Ohio Medical University USA
REVIEW RETURNED	12-May-2020

GENERAL COMMENTS	In my opinion, this study suffers from the following flaws: There is no comparison group used in the planned analysis. This would be a far stronger design. It would also allow for the potential differential effect of the predictors on the outcomes. Comparison to other non SMI databases needs to somehow match sample characteristics. Most importantly socioeconomic status differences need to be adjusted in that this too is a risk factor for poor health. This by the way should also be controlled for in the analysis. Some consideration of the effects of social isolation and environmental conditions should have been considered. It is not clear how the random sampling will occur. The measurement is unclear. How can health behaviors predict health outcomes without time frame considerations? Seems like mortality data needs to be used in a study that was designed to address pre-mature mortality. Perhaps a more careful reading of that literature would have been useful. In sum, I do not see why this should be published. I found nothing innovative here and am therefore skeptical that the results will be new, interesting or useful in addressing this important issue.
---

VERSION 1 – AUTHOR RESPONSE

Reviewer 1:	
Comment	Response
1. This study tackles a relevant and timely need for research and focuses on a population traditionally neglected. The objectives are well aligned with the background and although clear they are still a bit broad and additional specifications in this section and through the protocol would be helpful to promote reproducible research.	We are grateful to the reviewer for her positive comments and the time spent reviewing the paper. We hope that the changes outlined in response to the comments below help to provide more clarity and specificity. In the section addressing the objectives, we have included additional information; lines 110-120: “Objectives  1. Determine the prevalence of physical health conditions (i.e. type 2 diabetes, hypertension, heart disease, obesity and infectious diseases) and health-risk behaviours (i.e. diet, physical activity, alcohol and tobacco use) and compare these findings with those reported in the general population. 2. Determine the association between physical health conditions, health-risk behaviours, health-related quality of life and various demographic, behavioural, cognitive, psychological and social

	variables.  3. Identify health-service utilization and the type of lifestyle advice that has been offered for this SMI population. 4. Support clinical trial readiness by providing evidence-based findings regarding outcome measures and procedures relevant for clinical trials design. “ We have also included more information on the specifics of the survey design throughout the manuscript (see track changes). We have made an explicit commitment to share all materials and syntaxes with the public in the section on “Dissemination”: lines 333-335 “An open access repository of all materials will be built that covers all materials used in this survey as well as statistical syntax used by the team for their publications.” We have deleted the final sentence in the “Statistical Analysis” section since it mentioned additional/secondary analyses that are not directly related to the objectives and distract from the main purpose of the survey. We deleted: lines 301 to 303 “Additional analyses using network models⁴² and psychometric approaches will investigate comorbidity patterns (within and across physical and mental symptoms) as well as the structure of severe mental illness symptoms⁴³⁻⁴⁵”
 2. Design, settings and population: It would be helpful if the authors could specify which diagnostic criterion are they using to define their cohort. For example, it would be helpful to have the specific ICD codes for the SMI diagnosis. Moreover, age range and other potential socio-demographic factors for inclusion and exclusion would be interesting to have. 	We have matched the SMI diagnosis included in the study with the specific ICD codes and we have compiled all the inclusion and exclusion criteria for the population in the section of design settings and population. We have only excluded people who lack capacity, to be able to collect a sample of people with SMI as representative as possible; lines 154-163: “Adults (18 years and over) with a clinical diagnosis of SMI (i.e. (defined by the schizophrenialInternational Classification of Disease 10th revision (ICD-1023) as schizophrenia, schizotypal and delusional disorders (F20-F29); bipolar affective disorder (F30, F31); and depression with psychosis (F32.3, F33.3))that are able to provide informed consent as assessed by the treating clinician will be recruited. Patients who lack capacity to provide consent or are unable to complete study questionnaires will be excluded (figure 1). The data will be collected from June 2019 until September 2020.”
 3. For the sample size, it seems that there would be a number of participants by country. Are the authors going to use a stratified sampling strategy? As within 	We are using a by-country stratified sampling strategy, with a multi-stage sampling strategy at two of the sites to be able to have a sample containing 80% of outpatients and 20% of

institutions there will be also some randomization, would they follow a multi-stage strategy?	inpatients. We randomly selected patients and have now specified the procedures followed in each of the sites; line 144 “ (n=1500 per country)” and lines 150-158: “ At NIMHH and NIMHANS patients will be selected randomly using random number tables generated centrally to determine for outpatients, which patient to approach on a given day, and for inpatients by randomly selecting beds. Recruitment is separated by in- vs outpatients, resulting in a proportion of 80% outpatients and 20% inpatients (which has been reported as the “usual” proportion of in- and outpatients). Due to the lower volume of patients and outpatients reaching the centre outside working hours at the IOP, all patients attending the IOP during the study duration will be invited to participate in the survey. .”
4. With regards to the data collection, what is the plan to address the variability of dialects in countries as Pakistan and inequalities derived from urban vs rural areas, or potential stigma related to the topic.	The researchers in each site include males and females anticipating that some participants would not feel comfortable answering some of the survey questions to an interviewer from the opposite gender as well as to take the physical measurements (i.e. waist circumference, weight and height). We avoided the use of translators by 1) translating the survey to the most common languages and using translated validated versions of the tools in local languages when available; 2) recruiting researchers that speak multiple dialects. These points are now reported in lines 193-196: “The survey has been translated into the most common languages in the countries (Urdu in Pakistan, Bangla in Bangladesh and Hindi, Kannada, Tamil and Telugu for India). The team of interviewers in each county will include males and females and will include researchers that speak regional dialects, which is consistent with usual clinical practice in these settings²⁵.”
5. Some of these interviews might will need a local translator which can compromise confidentiality and the representativity of the sample. Is there any contingency plan for this? Also there could be some response bias related to the gender of interviewers or respondents (e.g., women will be less likely to respond some questions to male interviewers).	
6. Overall, all the information collected seem to be aligned with the objectives of the study but I find that two potential key elements are missed or not reported -	We are collecting information on these elements, and we have added this information; lines 206-209: “the STEPs demographic module will be used to

medication and socio-demographics (e.g., neighbourhood level, social disadvantage)- which have been consistently found to be relevant confounders in the associations that the authors are interested.	collect information on age, sex, education, marital status, employment, household assets and income. Mental health: We will collect information relevant to the SMI diagnosis of each participant, including duration of the mental illness, any treatment or medication and duration of the treatment and/or medication. The 9-item depression module (PHQ-9) will be used to measure the severity of depressive symptoms and the generalized anxiety disorder 7 item scale (GAD-7) for the severity of anxiety symptoms. .”
---	--

Reviewer 2:	
Comment	Response
1. In my opinion, this study suffers from the following flaws: There is no comparison group used in the planned analysis. This would be a far stronger design. It would also allow for the potential differential effect of the predictors on the outcomes.	The authors would like to thank the reviewer for taking the time to read the manuscript and his important comments. The main objective of the survey is to estimate prevalence of physical conditions and health risk behaviors in the SMI population from South Asia, where this information is lacking. There have been a number of publications highlighting the need for such studies (The, 2018 #48) Making more of multimorbidity: an emerging priority and (MacMahon, 2018 #49) https://acmedsci.ac.